# Efficacy and Safety of Holmium Laser Lithotripsy Under Local Anesthesia in the Treatment of Urethral Stones in Elderly Male Patients

**DOI:** 10.3390/healthcare8020150

**Published:** 2020-06-01

**Authors:** Dogan Atılgan, Engin Kölükçü, Fatih Fırat, Vildan Kölükçü

**Affiliations:** 1Department of Urology, Faculty of Medicine, Gaziosmanpasa University, Tokat 60100, Turkey; engin.kolukcu@gop.edu.tr; 2Department of Urology, Tokat State Hospital, Tokat 60100, Turkey; ffrat60@yahoo.com; 3Department of Anesthesia and Reanimation, Zile State Hospital, Tokat 60400, Turkey; vildanyaman@gmail.com

**Keywords:** elderly male, urethral stone, holmium laser, lithotripsy, local anesthesia

## Abstract

The elderly population has been increasing significantly in our century. In our study, it was aimed to analyze the treatment results of elderly male patients who underwent holmium laser lithotripsy (HLL) for urethral stones under local anesthesia. We evaluated a total of 31 male patients, aged ≥65 years, diagnosed with urethral stones and treated with HLL under local anesthesia. We noted the demographic data and visual pain scores (VAS) of the patients and the duration of the operation and hospital stay. Our analysis involved both the success rates of the surgical procedure and the complication rates according to the modified Clavien classification. In addition. we determined the patients’ preoperative clinical status using the Charlson comorbidity index (CCI). The mean age of the patients was 71.65 ± 8.19 years. Acute urinary retention was the most common complaint (45.2%). Their mean scores were 7.68 ± 2.53 according to CCI. The average operation time was 15.48 ± 5.22 min and the VAS was 2.03 ± 1.08. All patients were stone-free and there was a marked improvement in their symptoms None of them stayed in the hospital for more than one day. We did not observe any Grade 3 or higher complications. In light of the data obtained in our study, we concluded that HLL is an effective and reliable method to treat urethral stones under local anesthesia in elderly male patients.

## 1. Introduction

Aging is defined by the World Health Organization as “the decreased ability to adapt to environmental factors” and individuals aged ≥65 are called the elderly. Current technological developments related to public health have led to an increase the elderly population. Comprehensive epidemiological studies conducted in our geography estimate that—according to the 2000 data—the rate of the elderly population increased to 5.7%, and this rate will reach 21.7% in 2050. Similarly, while the life expectancy at the beginning of this century was 70.5 years, it is predicted that this period will reach 78.5 years in 2050 [1]. Many changes occur in the functions of the systems with aging, such as regression in the system functions, a decrease in organ reserves, weakening in hemostatic control, response to stress factors and adaptation to the environment. This makes the elderly more susceptible and vulnerable to pathologic clinical pictures. All this causes higher mortality and morbidity rates due to the systemic effects of both general and spinal anesthesia in older individuals. Numerous large-scale studies in this context report that surgical morbidity shows a linear increase with age [2,3].

Urinary stone disease account for the third most common clinical picture following prostate hyperplasia and infective pathologies in urology. However, urethral stones are extremely rare. Clinical studies report that less than 1% of the stones occurring in the urinary system are detected in the urethra. The treatment approaches in urethral stones vary depending on many factors such as the clinician’s experience, the stone’s size and location, the anatomic structure of the urethra and the patient’s clinical condition. Many different methods are used in the treatment such as delivery with lidocaine, meatotomy, extracorporeal shock waves and transurethral lithotripsy However, there is no detailed information about the diagnosis and treatment algorithms compared with other urinary stone disease [4]. We have observed that holmium laser lithotripsy (HLL) use under local anesthesia in the treatment of urethral stones in elderly male patients has not been analyzed previously in the literature. We discuss the results of HLL under local anesthesia in the treatment of urethral stones in male patients aged ≥65 years.

## 2. Material and Method

### 2.1. Patients

We evaluated 31 male patients aged ≥65 years who underwent HLL under local anesthesia for urethral stones between January 2016 and September 2019. The pre-intervention evaluation involved the medical history form, detailed physical examination findings, complete urinalysis, urine culture, routine biochemical and hematological analyses, direct urinary system graph and all non-contrast abdominal tomography, all recorded by a physician. Their clinical conditions were scored using the Charlson comorbidity index [5]. International Prostate Symptom Score (IPSS), Quality of Life (QoL), post micturition residue (PMR) and uroflowmetry measurement (UFM) values were evaluated at the time of admission, except for those who presented with acute urinary retention. Urethral anatomy, possible etiological factors and stone sizes and localizations were recorded by cystourethroscopy. We performed dilatation in patients with urethral stricture. The stones were fragmented with HLL. We analyzed the mean energy levels used for stone fragmentation, the stone-free rates, the duration of the procedure and the length of hospital stay. The patients’ pain levels and the complications of the procedure were evaluated according to the visual analog scale (VAS) and the modified Clavien classification system, respectively [6,7]. After the procedure, the patients were discharged and at the end of the first month all of them were called for control visit and IPSS, QoL, PMR and UFM evaluations. In addition, we recorded the results regarding the stone composition.

### 2.2. Inclusion and Exclusion Criteria

The study included male patients ≥65 years of age who were diagnosed with symptomatic urethral stones. Exclusion criteria were as follows: history of local anesthetic allergy, anatomic deformity that would preventing the lithotomy position, active urinary tract infection; those who did not give consent to the procedure to be performed under local anesthesia; cases where stones were observed in different localizations of the urinary system such as kidney, ureter and bladder; patients diagnosed with severe mental and psychological disorders.

### 2.3. Pain Measurement (Visual Analog Scale) 

VAS is a pain rating scale first used by Hayes and Patterson in 1921. Scores are based on self-reported measures of symptoms that are recorded with a single handwritten mark placed at one point along the length of a 10-cm line that represents a continuum between the two ends of the scale. There is “no pain” on the left end of the scale and the “worst pain” on the right end of it. The VAS score was assessed immediately after the surgery to assess the intraoperative pain [6].

### 2.4. Operative Technique

We examined the patients’ urine cultures before cystourethroscopy. If there was growth in the culture, we initiated an antimicrobial therapy according to their antibiograms. Endourological procedures were performed only after ensuring that the control urine cultures were clean. A percutaneous suprapubic cystostomy catheter was performed with urinary ultrasonography in those who presented with acute urinary retention. None of the patients had contraindications for percutaneous cystostomy such as bladder cancer, a severe bleeding disorder. The patients were informed in detail regarding the procedures to be followed and the local anesthesia. Considering the patients’ high mean age and comorbid conditions, we monitored their clinical conditions during surgical procedures. In this context, their vital signs and pain levels were monitored during the interventions by one anesthesiologist at least. We planned to terminate the procedure of the patients with abnormal vital signs and whose pain level was not suitable for continuing the surgical intervention. Pethidine HCI (Aldolan ampoule 100 mg/2 mL, Vem, Turkey) was administered to the patients as premedication 50 mg intramuscularly. In addition, a single dose of the first-generation cephalosporin was given proliferatively to each case an h before the procedure. Before the endourological intervention in the lithotomy position, we instilled 10-cc of 2% lidocaine gel and put a penile clamp to prevent it from leaving the urethra. After a waiting time of 15 min, we applied the same amount of lidocaine gel again into the urethra just before the flexible cystoscope entered the anterior urethra. All procedures were performed in a sterile environment and using a flexible cystoscopy (Karl Storz, El Segundo, CA, USA). When a urethral stricture was detected during the cystourethroscopy, we performed an s-curve dilatation with a guidewire. We used a 20 Fr S-Curve Dilator Set (Cook Medical, Bloomington, IN, USA) in the procedure. After reaching the urethral stones, we pushed the stones into the bladder to apply HLL within the bladder. When the stones could not be pushed into the bladder, HLL was applied in situ. We used the holmium: yttrium–aluminum–garnet laser device (Lisa Laser Sphinx 60 and Sphinx Jr, Germany) as the lithotripter. During the lithotripsy, we preferred two different probes, 272 µ and 365 µ, depending on the size of the stone. The laser energy was set at 0.5–1 J per between five and 20–40 Hz. We took the stone fragments out with foreign body forceps. Sterile 0.9%-NaCl solution was used for liquid irrigation during the procedure. 16Fr or 18 Fr urethral catheters were routinely inserted into each patient after the procedure. Catheter was removed first postoperative day. In those who underwent urethral dilatation, the urethral catheter was removed after three to seven days, depending on the severity of the stenosis.

### 2.5. Statistical Analyses

Descriptive analyses were performed to provide information on general characteristics of the study population. Quantitative data are shown as means ±SD or Med (1Q–3Q, the difference between the groups was examined by non-parametric method, the datas are shown with median and IQR 1 and 3). Independent sample *t-*test and Kruskal–Wallis tests were used to compare the continuous variables between the groups. Paired sample *t-*tests were used to calculate the difference between the pretest and posttest scores of variables. correlation analysis was used for examining the relationship between variables. A *p*-value < 0.05 was considered significant. Analyses were performed using commercial software (IBM SPSS Statistics 19, SPSS, Inc., an IBM Co., Somers, NY, USA).

### 2.6. Ethics Statement

The retrospective study was carried out following the principles of the Helsinki Declaration and with the approval of the local ethics committee. (Tokat Gaziosmanpasa University approval number: 20-KAEK-026).

## 3. Results

The age range of patients was 65–92 years and the mean age was 71.65 ± 8.19 years. The main findings were acute urinary retention in 14 (45.2%) patients, intermittent hematuria in 6 (19.3%), dysuria in 5 (16.1%), perineal pain in 3 (9.7%), weak urine stream in 2 (6.5%) and urethrorrhagia in one (3.2%). A total of 18 patients (58.1%) had positive urine cultures. The most common microorganism was *Escherichia coli* (50%) (Table 1). Of the patients in the study, 80.6% had at least one chronic disease and 54.8% had at least three. The most frequently observed pathologies were peptic ulcer, diabetes mellitus, coronary artery disease, dementia and cancers of the gastrointestinal tract. The average index scores of the patients were 7.68 ± 2.53 according to the Charlson comorbidity index.

The smallest stone size was 6 mm and the largest stone size was 20 mm. The mean stone size was 12.26 ± 3.93 mm. All but four patients had one stone. Three of these four patients had two stones located in the prostatic urethra and one had three stones located in the anterior urethra. Nineteen of the remaining 27 patients had a stone in the posterior urethra. In 23 of 31 patients (74.2%) stones were visualized on X-ray. However, only six stones (19.4%) could be palpated in the urethra in the genitourinary system examination. In addition. 24 patients (77.4%) had urethral stenosis. The most frequently accused etiological factor was transurethral prostate resection (37.5%) in these patients (Table 2). When we looked at the upper urinary systems of the patients, we encountered bilateral Grade 1 pelviectasis in seven patients and bilateral Grade 2 pelviectasis in two patients. The other patients did not have renal pelvis dilation.

The average operation time was recorded as 15.48 ± 5.22 min. Twenty four patients underwent urethral dilatation. Stones were only fragmented in four (12.9%) patients in situ. In the remaining patients, the stones were advanced into the bladder before the fragmentation. After the fragmentation, all stones were taken out with the help of a forceps. The mean VAS was calculated as 2.03 ± 1.08. There was no patient with severe pain during the procedure and no patients required analgesics after the procedure. Biochemical analysis of the stones revealed calcium oxalate (41.9%) to be the most frequently diagnosed stone compositions (Table 3).

Detailed analysis of the VAS showed that patients without urethral stricture were at significantly lower levels (*p* = 0.011) (Table 4). We used the Pearson correlation analysis to examine the relationship between VAS and urethral catheter duration and found a moderate, positive and significant relationship (*p* < 0.001) Similarly, we evaluated the relationship between VAS and stone size, length of operation, laser time and energy levels. We found a high level, positive and significant relationship between VAS and stone size (*p* < 0.001). Again, there was a moderate, positive and significant relationship between VAS and operation time (*p* = 0.004). However, there was no significant relationship between VAS and laser time and energy (*p* = 0.158 and *p* = 0.445, respectively) (Table 5). There was also no statistically significant relationship between VAS and stone composition (*p* = 0.997) (Table 6).

The preoperative IPSS and QoL values of the patients were 19.29 ± 7.28 and 4 ± 1.32, respectively (except for patients presented with acute urinary retention). Similarly, PMR and UFM results were evaluated. Preoperative PMR levels were 55.59 ± 28.39 cc. Preoperative UFM parameters were as follows: the mean maximum flow rate was 10.6 ± 4.04 mL/s, the mean flow rate was 5.55 ± 2.72 mL/s and the mean discharge time was 67.59 ± 11.12 s. All patients were called for control in the first postoperative month and their IPSS and QoL values were re-analyzed: 11.71 ± 5.03 and 1.65 ± 1.05, respectively. The PMR and UFM levels of all cases were noted. The PMRs were calculated as 25.65 ± 14.76 cc. When the UFM results were analyzed, the mean maximum flow rate of the parameters was 12.99 ± 3.94 mL/s, the mean flow rate was 7.49 ± 2.18 mL/s, and the mean discharge time was 41.42 ± 11.38 s. When the results of the patients who were presented with acute urinary retention and not included in the preoperative analysis were ignored, there was a statistically significant improvement in IPSS, QoL, PMR and all UFM parameters. (*p* < 000.1).

No procedure was interrupted due to pain or complications that arose during the endourological intervention and all patients were completely free of stones. We observed hematuria that did not require blood transfusion and lasted <24 h in seven patients, urethrorrhagia in two patients and urinary tract infection in one patient. None of the patients with complications required hospitalization. No patient had a urethral stricture secondary to the procedure. When postoperative complications were evaluated according to the modified Clavien classification, nine patients had Grade 1 and one patient had Grade 2 complications. However, no patients had any complications of Grade 3 and above that would cause mortality or morbidity according to the modified Clavien classification. All patients were discharged on the day of the operation. Their urethral catheters were removed during their follow-up period and under outpatient conditions.

## 4. Discussion

Urinary stone disease is among the oldest known disease in human history and have a history dating back to 4000 years BC. The incidence varies depending on many factors such as genetics, geographical region, climate conditions, sociocultural conditions and gender. The lifetime symptomatic urinary stone disease prevalence is approximately 13% in men and 7% in women. There was a significant increase in patients diagnosed with urinary stone disease in connection with the deteriorating eating habits and sedentary lifestyles of the communities in recent years [8]. Urethral stones are the rarest among all urinary tract stones and their clinical reflections vary widely. Patients can present to clinics with many different symptoms such as acute urinary retention, weak stream, frequent urination, hematuria, urethrorrhagia, dysuria, a mass in the penis and pain in the penile, rectal or perineal region [4,9]. Another reflection of urethral stones is urinary tract infections. These infections can occur with recurrent episodes of cystitis or urethritis and can cause life-threatening clinical conditions such as sepsis and penile gangrene [9]. In many clinical trials, the most common complaint of urethra stones was reported to be acute urinary system retention in accordance with our results [4]. On the other hand, urinary tract infection was detected in 58.1% of our cases.

X-ray, cystoscopy, computed tomography, magnetic resonance imaging and urethral physical examination are usually used to confirm the presence of urethral calculus. The most reliable method for the diagnosis is an endourological evaluation. The physical examination area is very limited. Previous studies report that only 8% of urethral stones can be palpated in the urethra [10]. However, as in other localizations of the urinary system, a significant part of the stones in the urethra contain calcium in their biochemical structure and it is seen as radiopaque stones with direct radiography with rates above 90%. The use of ultrasonography is very limited in urethral stones compared to other urinary tract stones [4]. In our study, 19.4% of the patients’ stones could be detected by physical examination and the stones could be observed by direct radiography in 74.2% of them. Although urethral stones may appear as primary urethral stones, secondary urethral stones that occur when the stones in the bladder, ureter or kidney migrate to the urethra are more common. In their series involving 56 cases, Koga et al. [11] reported that 32% of the patients had stones in a different localization of the urinary system. In another series of 36 cases conducted by Sharfi [12], 33% of the cases had associated urinary stone disease. A retrospective multicenter analysis by Jung et al. [13] stated that upper urinary tract stones and/or hydronephrosis increase the risk of urethral stones three time. Since our study was performed under local anesthesia, patients with stones localized only outside the urethra were excluded in the study. Therefore, we were unable to provide the stone rates found in other localizations of the urinary system.

A large part of the urethral stones leaves the urethra lumen spontaneously. Most of those that cannot be thrown spontaneously are localized in the posterior urethra. There are many etiological factors blamed to prevent the spontaneous passage of the stones. Previous endourological interventions, neurogenic bladder, infections, foreign bodies and anatomic disorders of the urethra are the most prominent of these factors [4,11]. In the literature, we observe that many treatment procedures are used in the treatment of urethral stones, especially minimally invasive methods. Endoscopic approaches have become the most preferred treatment modalities in the last ten years since they have given us the chance to evaluate the structure of the urethra as a whole and they are suitable for technical use in many clinics [10]. Nevertheless, open surgical approaches are still being applied in cases with large dimensions and especially in those related to urethral diverticula. However, they contain many complications such as high rates of infection, urethral stenosis, late wound healing and urinary fistulas. For this reason, open surgical procedure to extract urethral calculus was used as a last resort [10]. There are many factors effective in choosing the right surgical method such as the anatomic condition of the urethra, clinical presentations of the cases, general health conditions and the size and location of the stone [11,12].

With aging, a significant increase is observed in the frequency of drug use and rising comorbid pathologies. In addition, aged-related regression in body systems occurring as a natural result of aging causes clinicians to face great difficulties in the practice of both general and spinal anesthesia [14,15]. One of the most prominent of these difficulties is the ventilation of the patient during general anesthesia. Head and neck movements are also reduced with increased age. This limitation of motion is an independent determining factor in the prediction of difficult intubation. Another physical difficulty is oral and dental problems that are common in the elderly. It increases the risk of patients being exposed to dental trauma during laryngoscopy [16]. At an advanced age, the thermoregulatory control is impaired which may cause intraoperative hypothermia. This situation can disrupt cardiac nutrition and cause severe mortality results especially under unsuitable anesthesia conditions. With age, the speed and amount of diastolic filling decrease and many cardiac problems can be observed such as the loss of endothelial function, a decreased arterial elasticity and a decreased beta-receptor response. In addition to these conditions, cardiac rhythm disorders such as arterial fibrillation and tissue perfusion losses can be encountered during general and spinal anesthesia due to the deterioration of autoregulation mechanisms [17]. Aging also causes major changes in the respiratory system. The respiratory reserve and the gas exchange area of the alveoli decrease due to the loss of elasticity of the thorax structures and the increase in the anterior-posterior diameter of the chest. In addition. forced expiratory volume, ciliary activity and a decreased sensitivity of pharyngeal structures are observed. Depending on these conditions, elderly patients experience an increase in postoperative respiratory complications such as atelectasis, postoperative aspiration and bronchospasm [18]. In addition to these, it contributes to the high rates of anesthesia-related mortality and morbidity in changes in many different systems such as neurological, endocrinological, urinary and gastrointestinal organs. All this makes local anesthesia extremely important in geriatric patients.

An in-depth analysis of the previous studies revealed that most of the interventions performed only with local anesthesia in the treatment of urethral stones were applied for stones that can be diagnosed by inspecting the urethral mea. The two most common methods are delivering the stones with the help of a lidocaine gel and removing them with a direct thumb forceps. The physical manipulation of the stone and taking out of the urethra applying a lidocaine gel were first described by El-Sherif and El-Hafi. They found the success rate of this treatment approach as 77.8% by examining 18 patients with anterior urethral stones, who had urethral stones less than 10 mm and no history of urethral stenosis or urethral surgery [19]. In another study, Kilciler et al. [20] reported the success rate of two similar procedures in the treatment of anterior urethral stone as 88.2%. The biggest handicap of these approaches is that they can only be used in limited cases and the urethra is not considered as a whole. It should also be kept in mind that, when insisted on these manipulations, urethral stones with large dimensions, impacted into the lumen—and without a smooth surface—can cause damage to the urinary system mucosa. Another treatment approach applied to urethral stones with local anesthesia is extracorporeal shock wave lithotripsy (ESWL). The main mechanism of ESWL is based on the principle of ensuring fragmentation of the stone by turning the sound waves provided from the source outside the body into shock waves and directing them to the localized stone in the urinary system. However, its use in urethral stones is not as common as in other urinary stone disease. El-Sherif and Prasad [21] discussed the success of ESWL in 34 male cases with urethral stones between seven and 25 mm in size. Their study aimed to carry out ESWL by pushing urethral stones back to the bladder with the help of lidocaine gel and a foley probe. At the end of their clinical observations, they reported a success rate of 58.9% after one session of ESWL. Although the success rates were low for the first session, no patient required anesthesia or analgesia the same as in our study. Al-Ansari et al. [22] reported the overall success rate as 98.4% in their study they evaluated the success of ESWL in 64 patients diagnosed with urethral and bladder stones less than 25 mm, but without urethral stenosis. The same study reported that, a child patient needed general anesthesia and acute urinary retention developed in four patients after fragmentation. In our series, acute urinary retention did not develop in any patient after the procedure and we did not have a case that we performed general anesthesia.

In our century, lasers are used in urology clinics in a wide range of conditions from prostate hyperplasia to urethral strictures, cancer cases and stone disease. Lasers act with a combination of three mechanisms: photothermal, photomechanical and photochemical. holmium: yttrium–aluminum–garnet laser is the most commonly used laser type. This laser type has a wavelength of 2140 nm, shows rapid absorption in water and is safely applied worldwide in the treatment of urinary stone disease. However, there are very limited studies on the use of holmium laser under local anesthesia in stone disease [23]. Pai et al. [24] evaluated the use of holmium laser in ureter stones under local anesthesia. They discussed 85 cases and reported the success rate as 82%. Again, in the same study, they stated that all patients tolerated the procedure positively and no patient had a complication above Grade 2 according to the Clavien classification. Similar to our study, although the patients’ pain scores could not be documented, the procedure was not terminated due to pain either. Their wide-ranging study made a great contribution to the scientific world regarding the use of holmium laser under local anesthesia. Kara et al. [25] performed transurethral cystolithotripsy under local anesthesia in 13 cases with bladder stones of an average size of 3.6 cm and reported success rate as 100%. All patients had tolerable pain during the procedure and no clinical symptoms were encountered in 16.6 months of follow-up in their study. In another study D’Souza et al. [26] applied holmium laser cystolithotripsy under local anesthesia to 37 patients with bladder stones with a mean size of 2.1 cm. They stated that patients tolerated the procedure very well and that there were no intraoperative or postoperative major complications. They reported having received full results from their treatment with a short duration of hospitalization. The same study reported that the mean VAS score was 2.8 and the longest hospitalization time was five days. On the other hand, they did not detect any pathologic urological findings in the 6-month follow-up period. In our study, it was observed that recurrence was not detected according to our short follow-up data.

The limitations of this study were as follows; the data were analyzed retrospectively, the number of the cases was small and the long term results could not be documented, yet. Additionally, complete metabolic evaluation of the patients, such as 24-h urine analysis, could not be performed. Commentary upon the stone density of the patients during the preoperative period was also lacking due to technical limitations.

## 5. Conclusions

To our knowledge, this is the first study examining the use of the holmium laser for the treatment of urethral stones, especially in elderly patients who are more susceptible to high morbidity and mortality in interventional procedures under anesthesia. The data obtained in our study indicated the use of holmium laser has effective results in the elderly with high efficacy and low complication rates. We think that our data should be supported by randomized, large-scale and prospective studies to better guide the scientific world.

## Figures and Tables

**Table 1 healthcare-08-00150-t001:** Numbers and percentages of isolated microorganisms.

Pathogen	*n*	%
*Escherichia coli*	9	50
*Klebsiella pneumoniae*	4	22.2
*Enterococcus fecalis*	3	16.6
*Serratia marcescens*	1	5.6
*Candida albicans*	1	5.6

Urine culture was positive in 18 (58.1%) patients.

**Table 2 healthcare-08-00150-t002:** Etiological factors predicted in urethral stone formation.

Etiological Factors	*n*
History of transurethral prostate resection	9
History of endoscopic ureter stone surgery, endoscopic cystolithotomy and percutaneous nephrolithotomy	4
History of pelvic trauma	3
History of urethritis	3
Urethral diverticulum	1
Unknown	4

**Table 3 healthcare-08-00150-t003:** Chemical compositions of 31 stones.

Stone Composition	n
Calcium oxalate	13
Mixed calcium oxalate–calcium phosphate	7
Mixed Uric acid–calcium oxalate	5
Uric acid	3
Struvite	2
Cystine	1

**Table 4 healthcare-08-00150-t004:** Distribution of VAS scores according to urethral stricture.

	Urethral Stricture	*N*	Mean	Standard Deviation	*p*
VAS	Negative	7	1.1429	0.89974	**0.011**
	Positive	24	2.2917	0.99909	

VAS: visual analog scale; *p*: independent sample t-test. Bold: *p* < 0.05

**Table 5 healthcare-08-00150-t005:** Correlation between VAS scores and stone size, length of operation, laser time and energy levels.

VAS	Stone Size	Operation Time	Laser Time	Energy
r	0.716	0.500	0.260	−0.142
*p*	**<0.001**	**0.004**	0.158	0.445
*n*	31	31	31	31

VAS: visual analog scale; r: Pearson correlation coefficient.

**Table 6 healthcare-08-00150-t006:** Correlation between VAS and stone composition.

Stone Composition	VAS	*p*
calcium oxalate	3.00 (1.00–3.00)	0.997
calcium oxalate–calcium phosphate	2.00 (1.00–3.00)	
uric acid–calcium oxalate	2.00 (2.00–2.00)	
uric acid	2.00 (1.00–3.00)	
struvite	2.00 (2.00–2.00)	
cystine	2.00 (2.00–2.00)	

Data are shown as Med (1Q–3Q).

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
