# Peer review of "Efficacy and Safety of Holmium Laser Lithotripsy Under Local Anesthesia in the Treatment of Urethral Stones in Elderly Male Patients"

_healthcare, 2020, doi:10.3390/healthcare8020150_

Round 1
Reviewer 1 Report
In this well-written article, the authors report the outcomes of elderly male patients undergoing HLL for urethral stones under local anesthesia. The study design is appropriate for this kind of study. Personally, some issues should be explained more clearly throughout the paper:
As for intraoperative antibiotic prophylaxis to minimize the potential infectious complications following the procedure, what kind of protocol was applied?
Did any patients experience urethral strictures following the procedures?
It would be helpful if authors can provide detailed information about comorbidities of the patients but not just the average CCI scores given in the results section.
The authors should mention the limitations of their study.
There are some grammar and spelling mistakes to be edited.
Author Response
Dear editor,
Firstly, we would like to express our thanks for reviewing our article meticulously and guiding us with your constructive feedback.
We have made the following changes in line with your current suggestions.
We have detailed the antibiotic prophylaxis we applied in the twelfth sentence of the sixth paragraph of the material-method part.
We have added additional data regarding the content of our Charlson Comorbidity Index scores to the fifth sentence of the first paragraph of the result part.
We did not have any patients with urethral strictures during the procedures. We have detailed this in the fourth sentence of the sixth part of the results section.
We have added the limitations of the study to the last part of the discussion section.
We have fixed our grammatical mistakes.
Sincerely yours

Reviewer 2 Report
Congratulations for your manuscript. I feel that this is a useful article for clinical urologists, but I am afraid to say that it's not well written and needs improvement.
In general you must focus on the fact, that urethral stone HLL is feasible under local anaesthetic and avoid unnecessary details.
Abstract:
"We noted the patients' ages, symptoms, stone sizes and locations, biochemical compositions, etiological factors, operation times, visual pain scores (VAS), and complication rates according to the modified Clavien classification. In addition, we scored their preoperative clinical conditions using the Charlson Comorbidity Index. We analyzed all cases, except for those who presented with acute urinary retention, before and after the procedure using post-micturition residue urine volume (PMR), uroflowmetry (UFM) test results, international prostate symptom scores (IPSS), and quality of life (QoL) values. "
Too many details. Focus on CCI, VAS, Clavien and Hospital stay
A statistically significant improvement was achieved in 24 postoperative PMR, IPSS, and QoL values (p <000.1).
Incorrect p
Introduction:
2 first paragraphs should be 1 with less data.
Material and Methods:
Define inclusion and exclusion criteria clearly.
Tables 1 and 2 are not needed (citations are enough)
Please detailed description of VAS score and methodology.
Give titles to each paragraph eg Patients, Operative technique (+ catheter after procedure? when removed?..), Statistical...
Results:
The main findings: Main symptoms of presentation...
Make paragraphs smaller
More statitstics please eg correlation of laser time, energy with stone composition and VAS.
Discussion:
Smaller paragraphs... remove useless details. Correlate results in referred studies with your study.
Focus on VAS scores and compare the results with other urological studies studying similar procedures using VAS.
Author Response
Dear editor,
Firstly, we would like to express our thanks for reviewing our article meticulously and guiding us with your constructive feedback.
We have made the following changes in line with your current suggestions.
Abstract:
Following your suggestions, we reorganized the summary part towards the main target of the study.
In this context
‘We noted the patients’ ages, symptoms, stone sizes and locations, biochemical compositions, etiological factors, operation times, visual pain scores (VAS), and complication rates according to the modified Clavien classification. In addition, we scored their preoperative clinical conditions using the Charlson Comorbidity Index. We analyzed all cases, except for those who presented with acute urinary retention, before and after the procedure using post-micturition residue urine volume (PMR), uroflowmetry (UFM) test results, international prostate symptom scores (IPSS), and quality of life (QoL) values ‘
We removed this paragraph and rewrote it in a as described below.
We noted the demographic data and visual pain scores (VAS) of the patients, and the duration of the operation and hospital stay. Our analysis involved both the success rates of the surgical procedure and the complication rates according to the modified Clavien classification. In addition, we determined the patients’ preoperative clinical status using the Charlson Comorbidity Index (CCI).
We rearranged the section that expresses the results in the abstract.
In this context
‘The mean age of the patients was 71.65±8.19 years. Acute urinary retention was the most common complaint (45.2%). The average operation time was 15.48±5.22 minutes and the VAS was 2.03±1.08. All the patients were rendered stone free. None of the patients had complications of Grade III and above. A statistically significant improvement was achieved in postoperative PMR, IPSS, and QoL values (p <000.1). ‘
Removing this paragraph, we focused on the parameters of CCI, VAS, Clavien, and hospital stay and made it more specific as described below.
The mean age of the patients was 71.65±8.19 years. Acute urinary retention was the most common complaint (45.2%). Their mean scores were 7.68 ± 2.53 according to CCI. The average operation time was 15.48±5.22 minutes and the VAS was 2.03±1.08. All patients were stone-free and there was a marked improvement in their symptoms. None of them stayed in the hospital for more than one day. We did not observe any Grade 3 or higher complications in them.
Introduction
We changed the second sentence in the introduction section preserving its integrity. The removed and substituted sentences are as follows respectively:
‘The technological developments on public health in our century have led to higher life expectancy. As a result, the population distribution of societies has been changing significantly. Analyses in this context reveal a significant increase in the proportion of the elderly population in communities.’
‘Current technological developments related to public health have led to an increase both inlife expectancy and in the elderly population in communities.’
We have also simplified the following sentences in the last part of the second paragraph according to your suggestions:
‘A large-scale study by Turrentine et al. (2) evaluated 7696 surgical interventions for elderly patients in the United States. They reported a general mortality rate of 2.3% and a morbidity rate of 28%. The same study stated that mortality rates increased by 7% and morbidity rates up to 51% in over the age of 80. In a similar study, Oruc et al. (3) compared age with mortality and complication rates. They found the complication (35.6%) and mortality rates (23.28%) of patients aged >60 years higher than the complication (23.25%) and mortality rates (4.65%) of those <60 years of age.’
We replaced them with the following sentence instead:
‘Numerous large-scale studies in this context report that surgical morbidity shows a linear increase with age (2,3).’
In addition, two paragraphs have been converted into a single paragraph.
Material and Methods:
We reorganized the material and method section completely. In this context, we created the following subtitles: Patients, Inclusion and Exclusion Criteria, Charlson Comorbidity Index, Pain Measurement (Visual Analog Scale), Modified Clavien Classification System, Operative technique, Statistical Analyses and Ethics Statement.
In this context, we made changes in the references, subtracted Tables 1 and 2, and explained the Inclusion and Exclusion Criteria in detail.
We identified VAS and explained how it was implemented.
We explained the operation steps in a algorithm and added our data regarding the use of urethral catheters.
After adding our data, our statistical analysis approach has been updated.
We shared the ethics committee approval number.
Results:
We have corrected our grammar errors and shortened the paragraphs according to your suggestions.
The relationship between VAS scores and urethral stricture, urethral catheter duration, length of operation, stone composition, laser time and energy levels were evaluated in tables 4,5,6, and 7. In this context, we have added this data by adding paragraph 4.
Data regarding the length of hospital stay of our cases are presented in the last part of the results section.
Discussion:
We have shortened our paragraphs preserving their integrity.
The results of our study are correlated with other studies.
While examining other study results, we have focused on the VAS.
We have also corrected our grammatical mistakes.
We have added the following sentence at the end of the first paragraph
‘On the other hand, urinary tract infection was detected in 58.1% of our cases.’
We have removed the last sentence in the third paragraph, which is the following: ‘However, the need, duration, and type of anesthesia are also critical in determining the treatment strategies for elderly patients.’
We have also removed the following sentence in the fourth paragraph: ‘The use of pharmacological agents in elderly individuals is a condition that we encounter quite frequently. According to the research done by Nguyen et al. (14) in 2006, the elderly consume 45% of all medications in the UK and 33% in the USA. In addition, it has been determined that the elderly living in nursing homes use more medications and experience drug side effects more commonly than those living in the community. A large-scale study in the UK stated that 20% of the individuals >70 years of age use five or more medications (14). Clinicians face great difficulties at every stage of general or spinal anesthesia due to comorbid conditions and increased medication use, as well as changes that occur in almost all organisms with aging’.
We have replaced it with the following sentence: With aging, a significant increase is observed in the frequency of drug use and rising comorbid pathologies. In addition aged-related regression in body systems occurings as a natural result of aging causes clinicians to face great difficulties in the practice of both general and spinal anesthesia (14,15).
We have also revised these sentences in the last part of the same paragraph: ‘Residual volume decreases by 8-10% and the functional capacity by 1-3% every ten years. Besides, forced expiratory volume, ciliary activity, and a decreased sensitivity of pharyngeal structures are observed. Taken together, elderly patients are prone to the development of postoperative atelectasis. A large-scale study monitored the postoperative clinical course of patients aged >80 years and reported that 10.2% of the cases had pulmonary complications such as postoperative aspiration, bronchospasm, hypoxemia, pneumothorax, respiratory failure requiring treatment with positive pressure, and atelectasis (18). In addition to these, it contributes to the high rates of anesthesia-related mortality and morbidity in changes in many different systems such as neurological, endocrinological, urinary, and gastrointestinal organs. All this makes local anesthesia extremely important in geriatric patients’
We have replaced them with the following: Besides, forced expiratory volume, ciliary activity, and a decreased sensitivity of pharyngeal structures are observed. Depending on these conditions, elderly patients experience an increase in postoperative respiratory complications such as atelectasis, postoperative aspiration and bronchospasm (18). In addition to these, it contributes to the high rates of anesthesia-related mortality and morbidity in changes in many different systems such as neurological, endocrinological, urinary and gastrointestinal organs. All this makes local anesthesia extremely important in geriatric patients.
We have revised our fifth paragraph and added the following sentences to the last part: Although the success rates were low for the first session, no patient required anesthesia or analgesia the same as in our study. Al-Ansari et al. (22) reported the overall success rate as 98.4% in their study where they evaluated the success of ESWL in 64 patients diagnosed with urethral and bladder stones less than 25 mm but without urethral stenosis. The same study reported that, a child patient needed general anesthesia and acute urinary retention developed in four patients after fragmentation. In our series, acute urinary retention did not develop in any patient after the procedure, and we did not have a case that we performed general anesthesia.
We have added the following sentence at the end of the seventh sentence of the sixth paragraph:
Similar to our study, although the patients’ pain scores could not be documented, the procedure was not terminated due to pain either. Their wide-ranging study made a great contribution to the scientific world regarding the use of holmium laser under local anesthesia.
Again, we have added the following sentence in the last part of the same paragraph: The same study, reported that the mean VAS score was 2.8 and the longest hospitalization time was five days. On the other hand, they did not detect any pathological urological findings in the 6-month follow-up period. In our study, it was observed that recurrence was not detected according to our short follow-up data.
We have added the limitations of the study to the last part of the discussion section.
Sincerely yours

Round 2
Reviewer 2 Report
Abstract
- Remove “in them”
- ADD we concluded that HLL is an effective and reliable method to treat urethral stones under local 24 anesthesia in elderly male patients.
Introduction
JUST ..led to an increase of elderly population.
Urinary stone disease no diseases
Patients
REMOVE “We noted their” eg IPSS… were evaluated at the time of…
Don’t’ like sentence “All patients were called for control 71 visits”
Exclusion criteria were as follows: history of local anesthetic allergy; anatomical deformity preventing lithotomy position; active urinary tract infection;…
What was the duration of the follow up of the patients?
No need for paragraphs Charlson C score and Clavien. Just the reference. Well known to all.
Operative technique
Remove …and so on…
EEMOVE were followed up for one day with a ureteral catheter…. Catheter was remove first postoperative day.
Statistical AnalysIs
]Two ???
Results
REPHRASE: In direct radiography, stones had a 170 diagnostic value in 23 patients (74.2%).
REMOVE “Patients' pain levels were 183 measured with the VAS and”
REMOVE “the linearity of”
REMOVE table 5 does not add any additional information (you mention that in the text)
Discussion
No plural …. Urinary stone disease is …among the oldest known diseases
Rephrase all paragraph “The limitations of our study were as follows: the data were analyzed retrospectively, the number of cases was low, and the long-term results could not be documented. We could not perform the patients' metabolic evaluations, such as a 24-hour urine analysis, completely. Due to technical limitations, we were unable to comment on the stone density in any of our patients during the preoperative period.” TRY to avoid “we”
Author Response
2020, May,21st
Responses to Reviewer #2:
We would like to thank the Reviewer #2 for his/her kind appreciation of our manuscript and kind suggestions. The responses to the recommendations are as follows;
Abstract
- The term “in them” in the last paragraph of the manuscript was removed. Additionally, the term “to treat urethral stones” was added to the sentence beginning with “ We concluded that…”
Introduction
- The sentence in the introduction section of the manuscript beginning with “Current technological ….. …” was revised according to the request of the reviewer.
- The term “urinary stone diseases” was changed to “urinary stone disease”
Patients
- The sentence beginning with “ We noted ….” in the patients section was omitted and revised as a sentence beginning with “ Internatioanl Prostate Symptom Score (IPSS) ……..”
- The sentence beginning with “ All patients were called…..” was changed to a sentence as “After the procedure the patients were discharged and at the end of the first month all of them were called for control visit and IPSS, QoL, PMR and UFM evaluations.”
- The sentence describing the exclusion criteria was revised as the sentence beginning with “ Exclusion criteria history of local ….”
- There is not a median follow up time for this patient group. They were operated and immediate postoperative VAS scores were recorded and discharged. At the end of the first month control evalution scores were obtained and statistical comparisons were performed.
- The sections beginning with “ Clavien Classification system …..” and “Charlson C score ……” was omitted from the manuscript.
Operative technique
- The term “and soon” in the operative technique section was removed. The sentence beginning with “Cases without any urethral pathology were followed up for one day with a ureteral catheter ……..” was revised as a sentence beginning with “ catheter was removed …..”
Statistical Analysis
- In statistical analysis the unneceessary term “two” was removed.
Results
- The sixth sentence in the second paragraph of the results section was revised as the sentence beginning with “ In 23 of 31 patients ……”
- The sentence beginning with “Patients pain levels …..” in the result section was removed. The term “the linearity of” was discarded, as well. Table 5 was removed from the manuscript, and the numeric values of the tables were rearranged.
Discussion
- The plural terms “….diseases” were changed with singular terms.
The paragraph beginning with “ The limitation of our study …….” was rephrased
